# Changes in care-seeking for common childhood illnesses in the context of Integrated Community Case Management (iCCM) program implementation in Benishangul Gumuz region of Ethiopia

**Samson Gebremedhin**[1]*, **Ayalew Astatkie**[2], **Hajira M. Amin**[3], **Abebe Teshome**[3], **Abebe Gebremariam**[3]

**1** School of Public Health, Addis Ababa University, Addis Ababa, Ethiopia, **2** School of Public Health, Hawassa University, Hawassa, Ethiopia, **3** Emory University, Ethiopia, Addis Ababa, Ethiopia

* samsongmgs@yahoo.com

## Abstract

### Background

Integrated Community Case Management (iCCM) is a strategy for promoting access of under-served populations to lifesaving treatments through extending case management of common childhood illnesses to trained frontline health workers. In Ethiopia iCCM is provided by health extension workers (HEWs) deployed at health posts. We evaluated the association between the implementation of iCCM program in Assosa Zuria zone, Benishangul Gumuz region and changes in care-seeking for common childhood illnesses.

### Methods

We conducted a pre-post study without control arm to evaluate the association of interest. The iCCM program that incorporated training, mentoring and supportive supervision of HEWs with community-based demand creation activities was implemented for two years (2017–18). Baseline, midline and endline surveys were completed approximately one year apart. Across the surveys, children aged 2–59 months (n = 1,848) who recently had cough, fever or diarrhea were included. Data were analysed using mixed-effects logistic regression model.

### Results

Over the two-year period, care-seeking from any health facility and from health posts significantly increased by 10.7 and 17.4 percentage points (PP) from baseline levels of 64.5 and 34.1%, respectively ($p<0.001$). Care sought from health centres ($p = 0.420$) and public hospitals ($p = 0.129$) did not meaningfully change while proportion of caregivers who approached private ($p = 0.003$) and informal providers ($p<0.001$) declined. Caregivers who visited health posts for the treatment of diarrhea (19.2 PP, $p<0.001$), fever (15.5 PP,

**Data Availability Statement:** All relevant data are within the manuscript and its Supporting Information files.

**Funding:** The study was funded by UNICEF Ethiopia. The funder had no role in study design, data collection and analysis, decision to publish, or preparation of the manuscript.

**Competing interests:** HMA, AT and AG worked at Emory University (Ethiopia) that implemented the iCCM program. SG and AY received consultancy fees for evaluating the implementation of the program. We declare that the conflict of interest declared did not alter our adherence to all PLOS one policies.

$p<0.001$), cough (17.8 PP, $p<0.001$) and cough with respiratory difficulty (17.3 PP, $p =$ 0.038) significantly increased. After accounting for extraneous variables, we observed that care-seeking from iCCM providers was almost doubled (adjusted odds ratio = 2.32: 95% confidence interval; 1.88–2.86) over the period.

## Conclusion

iCCM implementation was associated with a meaningful shift in care-seeking to health posts.

## Background

Over the past three decades Ethiopia has made substantial progress in promoting child survival and achieved the Millennium Development Goal—4 target [1]. Between 1990 and 2018, under-five mortality rate (U5MR) has fallen by three-fourths from 206 to 55 deaths per 1,000 live births and nearly three million deaths were averted [1,2]. In line with the target set out in the Sustainable Development Goals (SDGs), Ethiopia is working towards reducing the U5MR to 25 deaths per 1,000 live births by 2030 [3].

Yet, annually an estimated 190,000 childhood deaths occur in Ethiopia largely due to manageable causes including pneumonia, diarrhea, malaria and newborn conditions [2,4,5]. Furthermore, the recent national mortality reduction has not been evenly observed across the regions of Ethiopia. In 2016 the average U5MR in the four emerging regions (Afar, Benishangul-Gumuz, Gambella, and Somali) was 102 deaths per 1000 live births, in contrast to 77 in the other four major regions [6]. Especially in the remote region of Benishangul Gumuz, the U5MR (98 deaths per 1,000 live births) stood at the second highest in the country and children from the region are twice as likely to die in the first five years of age as compared to those born elsewhere in Ethiopia [6,7].

Integrated Community Case Management (iCCM) is a strategy for increasing access to life-saving treatments for common childhood illnesses, through extending case management to trained, supported and supervised frontline health workers [8,9]. Experience from low- and middle-income countries suggested that frontline health workers can deliver quality community-based care to sick children in settings where access to formal health service is limited [10,11]. Since 2012, the World Health Organization (WHO) and the United Nations Children's Fund (UNICEF) have advocated the strategy and several countries have followed suit [8]. iCCM improves access to lifesaving treatment in hard-to-reach communities [10], provides correct treatment for malaria, diarrhea and uncomplicated pneumonia [11] and reduces child mortality [8,12,13] at an acceptable cost [14].

In Ethiopia, the implementation of iCCM program via the Heath Extension Program (HEP) was adopted in 2010 and scaled up to all regions [15,16]. The program is primarily provided through 32,000 trained Health Extension Workers (HEWs) deployed at 17,000 health posts and as of 2017, 95% of the health posts were providing the service [17]. Nonetheless, the coverage and quality of iCCM remains a concern due to several challenges including attrition of HEWs, shortage of medical supplies, limited demand for the service and lack of quality support to HEWs [15,16,18,19]. The iCCM is especially rudimentary in the emerging regions where the HEP platform is weaker [20,21].

Though iCCM is meant to improve access to care, utilization of the service provided by frontline health workers remains unsatisfactory [15,16]. A multicountry study that included 42

national surveys from sub-Saharan Africa and South Asia reported that community health workers were not the primary source of care for common childhood ailments [22]. In Ethiopia, in 2016 only 7.2% and 2.7% of children with diarrhea and fever, respectively, were managed by HEWs [6]. A study in Oromiya, Ethiopia also concluded that iCCM did not reduce child mortality due to low uptake of the service [23].

Even though the iCCM program started to be implemented in Ethiopia earlier, the program had been rudimentary in Benishangul Gumuz region until 2017. In 2017 Emory University (Ethiopia) and UNICEF in collaboration with Benishangul Gumuz Regional Health Bureau (RHB), took a two-year initiative to revitalized the iCCM program in Asossa zone, Benishangul Gumuz region. The program integrated training, mentoring and supportive supervision of HEWs with intensive community-based demand creation activities. This study presents the association between the implementation of the program and changes in care-seeking from different sources including the iCCM providers.

## Methods and materials

### Study design

Pre-posttest design without control group was applied to evaluate changes in care-seeking for common childhood illnesses (diarrhea, fever and cough) secondary to an iCCM program implemented in Assosa zone over two-year period (2017–18). Baseline survey was conducted ahead of the program in January 2017; and midline and endline surveys completed in January 2018 and January 2019.

### Study setting

The study was conducted in Assosa zone, one of the three zones of Benishangul Gumuz region of Ethiopia. Benishangul Gumuz is among the four emerging regions of Ethiopia having low socio-economic status, limited access to social services and weak human resources to implement development programs including health services. Assosa town, the regional and zonal capital, is located approximately 700 km northwest of the national capital Addis Ababa. In 2017 Assosa zone had 360,000 inhabitants, of whom 86% were rural dwellers [24]. Administratively, the area is divided into seven districts. The main sources of livelihood are subsistence mixed agriculture and artisan gold mining. Benishangul Gumuz region has the second highest poverty rates in Ethiopia and about a quarter of the population lives below the national poverty line [25]. Furthermore, road density and access to basic social services are extremely low.

According to the Ethiopia's three-tier healthcare system, primary care is provided by health posts (1 facility for 3,000 to 5,000 population), health centers (1 facility for 15,000 to 25,000 population) and primary hospitals (1 hospital for 60,000 to 100,000 population) [26]. At the time of the study, Assosa zone had 1 primary hospital, 7 health centers and 183 functional health posts. About 380 frontline health workers (mainly HEWs) were deployed at the health posts and provided preventive and basic curative services including the iCCM. In Benishangul Gumuz region including the study area, the HEP is relatively weak and health indicators are much lower than the national averages [6,17].

### Description of the intervention

The implementation of iCCM in the emerging regions of Ethiopia including Benishangul Gumuz region was started in 2015. However, at the ground level the program brought limited changes due to weak HEP platform, limited iCCM coverage, shortage of human resources for health and turnover of trained health workers. In 2017, Emory University in collaboration

with its partners revitalized the iCCM program in all the seven districts of Assosa zone. The Emory University's program included system strengthening and stronger community mobilization components. The program was directly involved in health system strengthening through training of frontline health workers and district health office managers, consolidating referral linkage between health posts and health centers, and instating quality improvement framework at various levels of the system.

The major programmatic activities the Emory University's program implemented included training of 182 HEWs on iCCM and 26 health professionals on Integrated Management of Newborn and Childhood Illnesses (IMNCI). The six-day iCCM training that was aligned with the standard WHO/UNICEF iCCM training protocol [27] addressed topics including clinical practice on identification and classification of signs and symptoms of common childhood illnesses and providing appropriate treatment, including pre-referral treatment, at health post level. Supervisory skill training was also provided to 42 HEW-supervisors. Furthermore, the program provided quality improvement training to 46 health professionals and the same was cascaded to 182 HEWs.

At community level, 885 health development army (HDA) members (network of community volunteers) and community leaders were trained on danger signs of childhood illness, community mobilization and prompt referral of sick children to health posts. Accordingly, the HDA members counseled caregivers and linked them with the primary health care system. Further, quarterly supportive supervision and monthly couching of HEWs were regularly implemented jointly by the RHB and Emory University's team. Multiple rounds of community festivals for iCCM demand creation were conducted on quarterly basis. Further, the festivals were used as opportunity for motivating community-level iCCM actors. Emory University also facilitated timely supply of iCCM commodities including medications to the health posts and worked towards improving referral linkage between health posts and health centers.

## Eligibility criteria

Mothers/caregivers from the seven districts of the of the zone having children 2–59 months of age with at least one of the illnesses in the past two weeks were eligible for the survey. Infants younger than two months of age and children from urban areas were excluded because they were not targeted by the iCCM program.

## Sample size calculation

The sample size for the survey was determined using the Cochran's single population proportion formula [28] assuming 39% expected prevalence of care-seeking from formal providers [29], 95% confidence level, 5% margin of error and design effect (DEFF) of 1.5. The DEFF is determined using the standard formula (DEEF = 1 + (cluster size-1)*intra-cluster correlation) specifying average cluster size of 23 and an intra-cluster correlation of 2% [30]. Based on this, 616 children were needed in each survey-round. Furthermore, using double population proportion formula, we assured that the sample size is adequate to detect 5 percentage points change in care-seeking between any two of the three surveys at 95% confidence level and 80% power.

## Sampling procedure

We used multistage cluster sampling approach for selecting the study participants. From each of the seven districts, 4 rural *kebeles*–the smallest administrative units in Ethiopia with approximately 1000 households, were randomly drawn. Then from each of the selected *kebeles*, 1 village (*got*) was chosen using simple random sampling (SRS) technique. In each selected village,

a rapid listing of eligible children was completed and sampling frames were developed. Ultimately from every village 23 eligible children got selected using SRS technique. With the intention of maximizing the sample size, study subjects who were not willing to take part in the study were replaced by randomly chosen eligible subjects from the same clusters. During each survey round, the same *kebele* and village were studied; however, study subjects were selected independently.

## Variables of the study

The two primary outcomes of the study were care-seeking (yes/no) from any health facility and care-seeking (yes/no) from health posts. Care-seeking was also determined for any of the individual conditions the child had in the past two weeks. Secondary outcomes were care-seeking from informal sources (e.g. traditional and religious healers) and caregivers' knowledge of childhood illness danger signs. The practice of consulting friends, family members or neighbors about the illness of the child was also considered as a secondary outcome. The factor of interest was implementation of the iCCM program (baseline, midline, endline). Control variables included distance of the household from the nearest health facility, household wealth index, maternal educational status, age and occupation, number of under-five children in the household and type of the primary caregiver (mothers vs other caregivers). The control variables were selected based on review of relevant literature [6,10,11].

## Data collection

Data were gathered by trained and experienced enumerators and supervisors from the primary caregivers of the children using pretested and interviewer administered questionnaire. The questionnaire had not been validated before but was used in a similar survey conducted in Ethiopia before. The questionnaire used in the survey is provided as a supporting file (S1 Table). The questionnaire was finalized in English, translated to Amharic language and administered to the respondents in their local languages. On top of socio-economic features, the questionnaire characterized the illness the child recently had and assessed the practice of care-seeking from different sources. Care-seeking practice was assessed based on the reports of the caregiver without reviewing any formal medical records. Occurrence of diarrhea was assessed by asking the caregiver whether the child had three or more loose stools per day in the reference period. For all children who reportedly had cough, the presence of concomitant difficulty of breathing and whether that was due to nasal congestion or chest problem, were explored.

Formal care providers were classified as public (health post, health center, public hospitals) or private (private clinic, hospital, charity clinics and drug vendors). Further, informal care-seeking was defined as care sought from informal practitioners (e.g. traditional or religious healers) or attempting traditional treatments at home.

The knowledge of the caregivers on thirteen danger signs of childhood illness [31] were assessed and a summated score (minimum and maximum possible scores of 0 and 13) was developed. Household wealth status was measured based on ownership of livestock, durable household assets, land size, materials used for house construction and access to electricity and improved drinking water source.

## Data management and analysis

We used STATA version 14 for data analysis. Descriptive data analysis was made using frequency distributions and measures of central tendency and dispersion. Weighted analysis was made using sampling weights and post-stratifications weights developed based on the population sizes of the districts. Changes in care-seeking and knowledge of danger signs of childhood

illness were compared across the surveys using chi-square for trend test. Household wealth index was developed using principal component analysis (PCA) as commonly done in national demographic and health surveys and classified into three tertiles: lower, middle and upper third. The association between the iCCM implementation (baseline, midline and endline) and formal care-seeking was evaluated using mixed effects multivariable logistic regression model. Random intercepts were set at district and *kebele* levels. Control variables were selected for adjustment based on statistical criteria. Initially, the comparability of the three surveys in selected basic socio-demographic characteristics was assessed using Pearson's chi-square test and significantly or marginally unbalanced variables ($p<0.1$) were statistically adjusted. The analyzed dataset is provided as a supporting table (S2 Table).

### Ethical considerations

The study was implemented in conformation with international ethical standards including the Helsinki Declaration. The work was approved by the institutional review board (IRB) of the Benishangul Gumuz Regional Health Bureau. The data were collected after taking informed verbal consent form the study subjects. Verbal, rather that written consent was used because significant proportion of the population in the area had no formal education. The same was approved by the ethics committee that cleared the protocol.

## Results

### Socio-demographic characteristics

Across the three surveys, a total of 1,848 interviews (616 per survey round) were made with caregivers of children with at least one of the three illnesses in the past two weeks. In nearly 99% of the cases data were collected from the mothers of the children and rarely other caregivers provided the information. The mean (±SD) age of the respondents was 28.5 (±7.7) years and about three-fourths (78.7%) were younger than 35 years. More than half (64.3%) had no formal education, 86.2% were housewives, and 94.4% were married/cohabiting. Regarding the characteristics index children, boys were slightly over-represented at 51.3%. The mean (±SD) age of the children was 27.1 (±13.0) months.

Table 1 summarizes the socio-demographic characteristics of the caregivers who took part in the three surveys. Statistically significant differences were observed across the surveys in terms of maternal educational status, household wealth index, maternal occupation, marital status, number of children under the age of five years in the household, age of the child and walking distance to the nearby health facility ($p<0.05$). Further, marginally insignificant difference ($p = 0.07$) was observed based on maternal age (Table 1).

### Morbidity pattern

Among 1,848 children represented in the surveys, 71.1% had cough, whereas 68.7% and 49.4% had fever and diarrhea, respectively. The proportion of children who experienced the three ailments were significantly different across the three surveys. Among 1848 children, a total of 3,450 ailments were reported in the preceding 2 weeks of the survey. One fifth (20.6%) reportedly had all of the three ailments; while 886 (47.9%) and 581 (31.4%) had two and one of the conditions, respectively (Table 2).

### Caregivers' knowledge of childhood illness danger signs

In the third-round survey nearly all of the caregivers (95.5%) were aware of at least one danger sign of childhood illness. The corresponding figure in the baseline was 90.1% and the

**Table 1. Socio-demographic characteristics of the study participants of the baseline, midline and endline surveys, Assosa zone, Ethiopia, 2017–2018.**

| Variables | Baseline (n = 616) | | Midline (n = 616) | | Endline (n = 616) | | All (n = 1,848) | | P-value |
|---|---|---|---|---|---|---|---|---|---|
| | Freq | % | Freq | % | Freq | % | Freq | % | |
| Type of the respondent | | | | | | | | | |
| Mother | 611 | 99.2 | 613 | 99.5 | 607 | 98.5 | 1831 | 99.1 | 0.190 |
| Other caregivers | 5 | 0.8 | 3 | 0.5 | 9 | 1.5 | 17 | 0.9 | |
| Respondent's age (years) | | | | | | | | | |
| 15–24 | 202 | 32.8 | 203 | 33.0 | 180 | 29.2 | 585 | 31.7 | 0.070 |
| 25–34 | 281 | 45.7 | 303 | 49.2 | 284 | 46.1 | 868 | 47.0 | |
| 35–44 | 132 | 21.5 | 110 | 17.9 | 152 | 24.7 | 394 | 21.3 | |
| 45 or above | 12 | 2.0 | 7 | 1.1 | 8 | 2.9 | 37 | 2.0 | |
| Maternal educational status | | | | | | | | | |
| No formal education | 429 | 69.7 | 374 | 60.8 | 385 | 62.6 | 1189 | 64.3 | 0.013* |
| Primary–first cycle | 42 | 6.8 | 50 | 8.1 | 53 | 8.6 | 145 | 7.9 | |
| Primary–second cycle | 99 | 16.1 | 147 | 23.9 | 137 | 22.3 | 383 | 20.7 | |
| Secondary or above | 46 | 7.5 | 45 | 7.2 | 40 | 6.5 | 131 | 7.1 | |
| Maternal occupation | | | | | | | | | |
| Housewife | 501 | 81.3 | 525 | 85.2 | 567 | 92.0 | 1593 | 86.2 | <0.001* |
| Traditional gold mining | 81 | 13.1 | 57 | 9.3 | 14 | 2.3 | 152 | 8.2 | |
| Others | 34 | 5.5 | 34 | 5.5 | 35 | 5.7 | 103 | 5.6 | |
| Marital status | | | | | | | | | |
| Married/Cohabiting | 573 | 93.0 | 579 | 94.0 | 593 | 96.3 | 1745 | 94.4 | 0.039* |
| Others | 43 | 7.0 | 37 | 6.0 | 23 | 3.7 | 103 | 5.6 | |
| Household size | | | | | | | | | |
| Less than five | 226 | 36.6 | 245 | 39.8 | 236 | 38.2 | 707 | 38.3 | 0.538 |
| Five or more | 390 | 63.4 | 371 | 60.2 | 380 | 61.8 | 1141 | 61.7 | |
| Sex of the baby | | | | | | | | | |
| Male | 329 | 53.4 | 296 | 48.1 | 323 | 52.4 | 948 | 51.3 | 0.134 |
| Female | 287 | 46.6 | 320 | 51.9 | 293 | 47.6 | 900 | 48.7 | |
| Age of the baby (months) | | | | | | | | | |
| 2–11 | 31 | 5.0 | 52 | 1.2 | 8.4 | 5.0 | 114 | 6.2 | 0.003* |
| 12–23 | 201 | 32.6 | 232 | 37.7 | 240 | 39.0 | 673 | 36.4 | |
| 24–59 | 384 | 62.3 | 332 | 53.9 | 344 | 55.9 | 1060 | 57.4 | |
| Number of children in the household | | | | | | | | | |
| One | 303 | 49.2 | 326 | 52.9 | 385 | 62.5 | 1014 | 54.9 | <0.001* |
| Two or more | 313 | 50.8 | 290 | 47.1 | 231 | 37.5 | 834 | 45.1 | |
| Wealth index | | | | | | | | | |
| Poor | 209 | 33.9 | 245 | 39.8 | 162 | 26.3 | 616 | 33.3 | <0.001* |
| Middle | 209 | 33.9 | 208 | 33.8 | 198 | 32.1 | 616 | 33.3 | |
| Rich | 198 | 32.1 | 163 | 26.5 | 256 | 41.5 | 616 | 33.4 | |
| One-way walking distance to the nearest health facility | | | | | | | | | |
| 30 minutes or less | 550 | 89.3 | 597 | 96.9 | 593 | 96.3 | 1740 | 94.2 | <0.001* |
| More than 30 minutes | 66 | 10.7 | 19 | 3.1 | 23 | 3.7 | 108 | 5.8 | |

Statistically significant difference at *p*-value of 0.05.

difference was statistically significant (*p*<0.001). Among thirteen danger signs we considered, the mean (± standard deviation) number of danger signs identified by the caregivers significantly increased from 2.43 (±1.25) in the baseline to 2.69 (±1.18), in the endline survey

**Table 2. Morbidity pattern of children 2–59 months in the preceding two weeks of the survey, Assosa zone, Ethiopia, 2017–2018.**

| Variables | Baseline (n = 616) | | Midline (n = 616) | | Endline (n = 616) | | All (n = 1,848) | | *P* value |
|---|---|---|---|---|---|---|---|---|---|
| | Freq | % | Freq | % | Freq | % | Freq | % | |
| Children who had | | | | | | | | | |
| Cough/breathing difficulty | 455 | 73.9 | 449 | 72.9 | 410 | 66.6 | 1314 | 71.1 | 0.009* |
| Fever | 422 | 68.5 | 465 | 75.5 | 382 | 62.0 | 1270 | 68.7 | <0.001* |
| Diarrhea | 331 | 53.7 | 286 | 46.4 | 296 | 48.1 | 913 | 49.4 | 0.027* |

Statistically significant difference at *p*-value of 0.05.

($p$<0.001). Across the surveys, significant linear increments were observed in the proportion of caregivers who considered fever, unable to drink or feed, measles, hypothermia and lethargy as danger signs ($p$<0.05) (Table 3).

## Care-seeking for common childhood illnesses

Table 3 summarizes the care-seeking for common childhood illness from different health facilities during the three survey rounds. Over the period, care-seeking from any formal provider and from health posts significantly increased by 10.7 and 17.4 percentage points (PP) from baseline levels of 64.5 and 34.1%, respectively ($p$<0.001). However, care-seeking from health centres ($p$ = 0.420) and public hospitals ($p$ = 0.129) remained unchanged while care-seeking from private providers significantly declined ($p$ = 0.003).

Results for individual medical conditions demonstrated a similar pattern. During iCCM implementation, care-seeking from health posts for treatment of diarrhea (19.2 PP, $p$<0.001), fever (15.5 PP, $p$<0.001), cough (17.8 PP, $p$<0.001) and cough with breathing difficulty (17.3 PP, $p$ = 0.038) significantly improved (Table 4).

**Table 3. Caregivers' knowledge of childhood illness danger signs, Assosa zone, Ethiopia, 2017–2018.**

| Variables | Baseline (n = 616) | | Midline (n = 616) | | Endline (n = 616) | | *p*-value[+] |
|---|---|---|---|---|---|---|---|
| | Freq | % | Freq | % | Freq | % | |
| % aware of at least one danger sign | 555 | 90.1 | 584 | 94.8 | 588 | 95.5 | <0.001* |
| Reported danger signs | | | | | | | |
| Fever | 523 | 84.9 | 546 | 88.6 | 562 | 91.1 | 0.001* |
| Unable to drink or feed | 226 | 36.7 | 226 | 36.7 | 302 | 49.0 | <0.001* |
| Fast/difficult breathing | 137 | 22.2 | 183 | 29.7 | 117 | 19.0 | 0.180 |
| Persistent vomiting | 175 | 28.4 | 181 | 29.4 | 162 | 26.3 | 0.614 |
| Persistent diarrhea or dysentery | 286 | 46.4 | 343 | 55.7 | 257 | 47.9 | 0.098 |
| Measles | 14 | 2.3 | 30 | 4.9 | 62 | 10.1 | <0.001* |
| Convulsion | 48 | 7.8 | 84 | 13.6 | 54 | 8.8 | 0.570 |
| Hypothermia | 15 | 2.4 | 31 | 5.0 | 47 | 7.6 | <0.001* |
| Lethargy | 24 | 3.9 | 15 | 2.4 | 45 | 7.3 | 0.004* |
| Sunken eye | 13 | 2.1 | 10 | 1.6 | 8 | 1.3 | 0.267 |
| Jaundice | 7 | 1.1 | 13 | 2.1 | 11 | 1.8 | 0.375 |
| Skin pinch going back slowly | 18 | 2.9 | 17 | 2.8 | 12 | 1.9 | 0.278 |
| Severe chest in-drawing | 12 | 1.9 | 19 | 3.1 | 17 | 2.8 | 0.371 |
| knowledge on danger signs (mean ± sd) | 2.43 (±1.25) | | 2.76 (±1.78) | | 2.69 (±1.18) | | <0.001* |

[+] Linear by linear chi-square test

* Statistically significant difference at 5% level of significance.

**Table 4. Care-seeking for common childhood illness Assosa zone, Ethiopia, 2017–18.**

| Type of ailments | Health seeking (%) for common childhood illness | | | | | | | | | | | | | | | | | | | | |
|---|---|---|---|---|---|---|---|---|---|---|---|---|---|---|---|---|---|---|---|---|---|
| | Any health facility | | | | Health post | | | | Health center | | | | Public hospital | | | | Private sector | | | |
| | BS | MS | ES | P-value | BS | MS | ES | P-value | BS | MS | ES | P-value | BS | MS | ES | P-value | BS | MS | ES | P-value |
| Cough, fever or diarrhea (n = 3,450) | 64.5 | 68.4 | 75.2 | <0.001* | 34.1 | 39.4 | 51.5 | <0.001* | 28.8 | 35.5 | 30.2 | 0.420 | 1.1 | 1.3 | 0.5 | 0.129 | 10.4 | 7.2 | 7.4 | 0.003* |
| Diarrhea (n = 904) | 73.1 | 77.4 | 81.3 | 0.015* | 37.9 | 44.9 | 57.1 | <0.001* | 33.9 | 39.6 | 34.0 | 0.947 | 1.2 | 1.4 | 0.3 | 0.280 | 11.0 | 7.8 | 5.8 | 0.018* |
| Fever (n = 1,249) | 63.1 | 67.8 | 74.0 | 0.001* | 33.7 | 38.8 | 49.2 | <0.001* | 25.9 | 34.9 | 28.4 | 0.416 | 1.0 | 0.9 | 0.3 | 0.236 | 10.8 | 7.2 | 7.3 | 0.750 |
| Cough (n = 1,271) | 59.6 | 63.2 | 71.8 | <0.001* | 31.7 | 36.5 | 49.5 | <0.001* | 27.7 | 33.6 | 29.2 | 0.604 | 1.1 | 1.6 | 0.7 | 0.631 | 9.8 | 6.8 | 7.8 | 0.290 |
| Cough with difficult breathing (n = 200) | 63.5 | 71.1 | 80.3 | 0.038* | 36.8 | 41.3 | 54.1 | 0.016* | 28.6 | 30.3 | 32.8 | 0.612 | 1.6 | 1.3 | 0.0 | 0.378 | 9.5 | 11.8 | 9.8 | 0.951 |

BS = Baseline survey; MS = Midline survey; ES = Endline survey

* Significant positive or negative trend at 5% level of significance.

The association between the iCCM implementation and care-seeking was further evaluated using multivariable mixed-effects logistic regression analysis. In the model adjusted for possible confounders including one-way walking distance to the nearest health facility, care-seeking from iCCM providers significantly increased by almost two folds (AOR = 2.32: 95% CI; 1.88–2.86) in the endline as compared to the baseline survey. However, no difference was observed between the first two surveys (Table 5).

Among respondents who did not seek care from any health facility, their underlying reasons were explored. The major reasons were: thinking the disease is not severe (56.2%), financial constraints (28.6%), child got sick very recently (12.8%) and being busy with household chores (6.0%). Other less frequently mentioned reasons were: health facility was closed (3.2%), transportation problem (2.7%) and underestimating the service quality at the nearby health facility (2.2%).

## iCCM and informal care-seeking

As described earlier, informal care-seeking was operationally defined as care sought from informal practitioners (e.g. traditional or religious healers) or attempting traditional treatments at home. In the baseline survey 5.1% of the caregivers reported such practices, however the figure significantly declined to 3.4 and 0.6% in the midline and endline surveys, respectively (p<0.001). In the multivariable model adjusted for possible confounders, informal care-seeking in endline was significantly reduced by 87% (AOR = 0.13: 95% CI; 0.06–0.29) taking

**Table 5. Association between iCCM program implementation and changes in care-seeking for common childhood illness from health posts in Assosa zone, Ethiopia, 2017–18.**

| Survey round | Odds ratio (95% CI) | |
|---|---|---|
| | Crude odds ratio | Adjusted odds ratio‡ |
| Baseline | 1 | 1 |
| Midline | 1.27 (1.06–1.53)* | 1.16 (0.96–1.41) |
| Endline | 2.55 (1.10–3.10)* | 2.32 (1.88–2.86)* |

* statistically significant association at 5% level of significance.

‡ adjusted for one-way waking distance to the nearest health facility, child's and caregivers age, maternal marital status, educational status and type of occupation, household wealth index, and type of ailment.

the baseline survey as the reference. The difference between the first two survey rounds was marginally insignificant (AOR = 0.65: 95% CI; 0.42–1.00) ($p$ = 0.052).

Similarly, in the baseline survey 29.7% of the caregivers consulted friends, neighbours or family members about the sickness the child. The corresponding figures were 28.1% and 20.6% in the midline and endline surveys, respectively and the decline was statistically significant ($p<0.001$). In the multivariable model adjusted for possible confounders, such practice was significantly declined by 35% (AOR = 0.65: 95% CI; 0.52–0.81) in endline as compared to the baseline survey. The difference between the first two surveys was insignificant (AOR = 0.95: 95% CI; 0.78–1.15).

## Discussion

The study found that implementation of iCCM program within the health extension program package was associated with a meaningful increase in care-seeking for common childhood illnesses, especially from iCCM providers–frontline workers at health posts. The iCCM implementation was also associated with a decline of informal care and care sought from private providers.

Over the project period, care-seeking from any health facility and from health posts significantly increased by 10.7 and 17.4 PP. Similar findings have also been documented elsewhere [10,32,33]. A study in Zambia documented increase in care sought for malaria and pneumonia from frontline health workers in areas where community health workers were trained and provided with essential iCCM supplies [32]. In Nigeria an iCCM program that incorporated demand creation activities successfully enhanced care-seeking for fever, diarrhea and fast breathing by 13–19 PPs [10]. Similarly, in Ghana and Uganda iCCM improved prompt care-seeking practices [32,33]. However, two studies in Ethiopia that compared care-seeking in iCCM implementing and control districts found no significant differences [20,34]. The findings may indicate that the effect of iCCM on care-seeking may depend on multiple contextual factors including intensity of demand creation activities.

Implementation of iCCM program in Assosa zone was also associated with a significant decline in care sought from private providers. A cluster randomized trial in Oromiya region, Ethiopia observed that in the first two years of introduction of the program, care-seeking from private facilities declined by 5 PP in the intervention and as compared to 2 PP in the control arm [23]. Similarly, in Jimma and West Hararghe zones of Ethiopia, a study reported that iCCM caused 66% reduction in the utilization of private providers [34]. The finding is likely to be due to shifting of care-seeking from private to iCCM providers.

The study found that informal care-seeking and the practice of consulting friends, neighbours or family members on sickness of children significantly declined during the iCCM implementation period. A randomized control trial in Oromiya, Ethiopia also concluded that care-seeking from informal sources, including traditional healers, shops and friends, decreased by 13 PP in iCCM implementing areas, in contrast to 8% in control districts [23]. This can also be considered as another reflection of shifting of care-seeking to frontline health workers as the result of the iCCM program. The decline in the popularity of friends, neighbours or family members as sources of health information might have resulted from the growing trust and confidence of the community on HEWs.

The present study also demonstrated that caregivers' knowledge of danger signs of childhood illness significantly increased during the implementation period suggesting that it might be among the pathways that led to improvement of care-seeking practice. The change in knowledge can be due to health messages disseminated by the iCCM actors through multiple channels including community festivals, interpersonal communication and the HDA network.

A study in Nigeria also reported that during the iCCM implementation caregivers who identi-fied three or more danger signs were significantly raised by 12 PP [10]. However, in Jimma and west Hararghe zones of Ethiopia no significant change on knowledge of danger signs, as well as care-seeking practice, secondary to the iCCM was observed [34].

The findings of the study should be interpreted in consideration of the following methodo-logical strengths and shortcomings. On a positive note, we surveyed large number of caregivers and monitored the implementation of the program over two-year period through three large-scale surveys. Conversely, as the study was not a controlled trial, maturation effect is possible and this might have caused us to overestimate the effect of the program. For instance, the observed improvements might be partially attributable to contextual and system-wide changes not directly related with the iCCM program. Furthermore, prior to the implementation of the program, rudimentary iCCM had already been in place in the area and this might underesti-mate the benefit of the program. Though we have attempted to statistically offset socio-demo-graphic variations observed among the surveys, distortion from unmeasured or residual confounders cannot be entirely excluded. Figures on care-seeking could have also been exag-gerated due to social desirability bias. Furthermore, infants 2–11 months were underrepre-sented (contributed to only 5–6% of the total children enrolled) which may limit the generalizability of the findings of the study. Furthermore, we only evaluated changes in care-seeking and did not look into other pertinent program dimensions including quality and impact on mortality of the iCCM program. Beyond training of frontline health workers, the rate of retention of skills had not been evaluated. In addition, we did not collect data on changes of medical supplies and IEC activities before and after the initiation of the program.

## Conclusion

The implementation of iCCM program having an inbuilt demand creation component was associated with a meaningful increase in care-seeking for common childhood illness, especially from health posts and decline in informal care and care sought from the private sector. How-ever, as the study was not a controlled trial the changes in care seeking can also be due to matu-ration effect.

## Supporting information

**S1 Table.**
(DOCX)

**S2 Table.**
(XLSX)

## Acknowledgments

The authors appreciate all the study participants, data collectors and supervisors for realizing the study.

## Author Contributions

**Conceptualization:** Hajira M. Amin, Abebe Teshome, Abebe Gebremariam.

**Data curation:** Samson Gebremedhin.

**Formal analysis:** Samson Gebremedhin, Ayalew Astatkie.

**Funding acquisition:** Abebe Gebremariam.

**Investigation:** Samson Gebremedhin, Ayalew Astatkie, Hajira M. Amin, Abebe Teshome.

**Methodology:** Samson Gebremedhin, Ayalew Astatkie, Hajira M. Amin.

**Project administration:** Abebe Gebremariam.

**Supervision:** Samson Gebremedhin, Ayalew Astatkie, Hajira M. Amin, Abebe Teshome, Abebe Gebremariam.

**Writing – original draft:** Samson Gebremedhin.

**Writing – review & editing:** Samson Gebremedhin, Ayalew Astatkie, Hajira M. Amin, Abebe Teshome, Abebe Gebremariam.

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
