## [Decision Letter · Decision Letter 0]

21 Sep 2020

PONE-D-20-26348

Changes in care-seeking for common childhood illnesses in the context of Integrated Community Case Management (iCCM) implementation in an emerging region of Ethiopia

PLOS ONE

Dear Dr. Gebremedhin,

Thank you for submitting your manuscript to PLOS ONE. After careful consideration, we feel that it has merit but does not fully meet PLOS ONE’s publication criteria as it currently stands. Therefore, we invite you to submit a revised version of the manuscript that addresses the points raised during the review process.<please by="" manuscript="" revised="" submit="" your="">

Please include the following items when submitting your revised manuscript:</please>

We look forward to receiving your revised manuscript.

Kind regards,

Khin Thet Wai, MBBS, MPH, MA (Population & Family Planning Resear

Academic Editor

PLOS ONE

Journal Requirements:

2. Please include additional information regarding the survey or questionnaire used in the study and ensure that you have provided sufficient details that others could replicate the analyses. For instance, 1)if you developed a questionnaire as part of this study and it is not under a copyright more restrictive than CC-BY, please include a copy, in both the original language and English, as Supporting Information, and 2) whether the questionnaire has been previously validated.

"HMA, AT and AH worked at Emory University (Ethiopia) that implemented the iCCM

program. SG and AY received consultancy fees for evaluating the implementation of

the program."

Additional Editor Comments (if provided):

This article highlights the role of trained health workers in improved community care seeking. Following issues are to be considered to strengthen readability and scientific integrity of the manuscript:

(1) It is essential to correct grammatical errors and typos.

(2) To clarify whether the same households in kebeles selected/recruited in three rounds of surveys and whether there was any replacement strategy used for vacant households.

(3) The authors need not change the use of verbal consent to written informed consent as suggested by the reviewer. Oral informed consent is acceptable for those with low literacy.

(4) In the discussion section, the possible impact of demand creation activities and uninterrupted medical supplies system should be added with appropriate citation rather than included in the results section as indicated by the reviewer:

" It will be good to give comparison of change of IEC activities before and after the programme, the medicine supply chain".

(5) If the positive findings are likely to be due to " maturation effect" please rewrite the conclusion.

Reviewers' comments:

Reviewer's Responses to Questions

**Comments to the Author**

1. Is the manuscript technically sound, and do the data support the conclusions?

Reviewer #1: Yes

Reviewer #2: Yes

2. Has the statistical analysis been performed appropriately and rigorously? 

Reviewer #1: No

Reviewer #2: Yes

3. Have the authors made all data underlying the findings in their manuscript fully available?

Reviewer #1: No

Reviewer #2: Yes

4. Is the manuscript presented in an intelligible fashion and written in standard English?

Reviewer #1: Yes

Reviewer #2: Yes

5. Review Comments to the Author

Reviewer #1: The authors have addressed an important issue of access to care near hone where referral is not possible . There are few issue which need attention.

Title: It is unclear why the authors have used the word ‘emerging” for region. There is no mention about registry of the trial , hope it has been registered .

Background : The ICCM programme was implemented in Ethopia in 2012 , but it is unclear what was the implementation status before the research team implemented the programme .

Methods : Authors have used a pre and post design and the aim is to evaluate the change in care seeking behaviours after implementation of the programme not the associations ?the objectives need to be clearly stated .

Intervention: The authors need to describe the intervention in more details , what was missing , what was put in place by the research team . How was health system strengthening done .

IEC : what IEC activities were done specific to ICCM programme , its frequency..

Supervision: More details about the supervision , what was done , what was the rate of retention of skills will be valuable to add

Sampling procedure : It is unclear how the households were selected for the three surveys , were the same kebeles selected .

Ethical consideration : Although the ethics committee has cleared the project , a witnessed written informed consent could have been obtained

Data Analysis: The rationale of selecting variables for adjustment has not been described . The authors need to give definition of the wealth index if that is consistent with the definition used in national surveys .

Tables and Results : It will be good to give a trial profile of the exclusions , the authors have not mentioned children with what illness were selected .

Table ! there is typographical error endline survey has been labelled as midline .

The programme was implemented for 2-59 months , however the mean age of children who were included in survey is 27 months, it is unclear whether the younger children did not participate or did not have any illness .

It will be good to give comparison of change of IEC activities before and after the programme, the medicine supply chain

Reviewer #2: The objective of this study is of great relevance to health of under 5 years old children residing in areas with limited access to facility-based health care provider. The paper is well-written and use appropriate study design and procedure. As the author mentioned, the major limitation is that there is no control arm in the study that could affect the empirical evidence of the effectiveness of intervention. There are only few comments which are shown below.

Line 84: There may be a typographical error, “A multicountry study that included ……”

Line 91: What is meaning of “revitalized the iCCM program …”? Is there previous history of implementing iCCM in the area? If yes, how is it different from previous iCCM program?

Line 148 & 150:What are references for design effect 1.5 and intra-cluster correlation 2%?

Line 168: The author mentioned that care-seeking was defined as any of the individual conditions the child had during two weeks. But did not explain clearly how to determine when two or more signs and symptoms (e.g. fever with cough and diarrhea) were occurred in the single time of illness. In my opinion, the definition of care-seeking practice should be whether they visited to health facility or health post for every single time of illness. Reporting number of care seeking practice based on individual signs and symptoms (S & S) would lead estimation bias because one or more S & S can be occurred during single time of illness.

Line 182: How the care-seeking practice was assessed? Is it record review or reported care-seeking practice by asking questionnaire?

Line 191: What is reference for 13 danger signs?

In Table 1, the 4th column heading should be “Endline”.

There is same typographical error in the paragraph interpreting table 4 and footnote of table 4, which should be “one way walking distance…”.

Overall, the results of this study will contribute towards increasing the access to treatment needs in under 5 years old children residing in areas with limited access to facility-based health care provider. I am grateful for being considered to review this manuscript and would gladly review any updated versions in the future.

6. PLOS authors have the option to publish the peer review history of their article (what does this mean?). If published, this will include your full peer review and any attached files.

Reviewer #1: No

Reviewer #2: No

---

## [Author Response · Author response to Decision Letter 0]

27 Sep 2020

Thank you for both of the reviewers and the editor for raising all these important points. We have tried our best to accommodate all of them. 

Journal Requirements:

Comment 1: When submitting your revision, you need you to address these additional requirements on file naming.

Response: Done. 

Comment 2: Please include additional information regarding the survey or questionnaire used in the study and ensure that you have provided sufficient details that others could replicate the analyses. For instance, 1)if you developed a questionnaire as part of this study and it is not under a copyright more restrictive than CC-BY, please include a copy, in both the original language and English, as Supporting Information, and 2) whether the questionnaire has been previously validated.

Response: The data collection tool is now given as the supporting table. Please note that the tool had not been validated before and we have stated the same in the “Data collection” sub-section. 

Comment 3: We note that you have indicated that data from this study are available upon request. PLOS only allows data to be available upon request if there are legal or ethical restrictions on sharing data publicly. If there are no restrictions, please upload the minimal anonymized data set necessary to replicate your study findings as either Supporting Information files or to a stable, public repository and provide us with the relevant URLs, DOIs, or accession numbers.

Response: We have now uploaded the minimal anonymized data as a supporting table. 

Comment 4: Thank you for stating the following in the Competing Interests section:

"HMA, AT and AH worked at Emory University (Ethiopia) that implemented the iCCM

program. SG and AY received consultancy fees for evaluating the implementation of

the program."

Please confirm that this does not alter your adherence to all PLOS ONE policies on sharing data and materials, by including the following statement: "This does not alter our adherence to PLOS ONE policies on sharing data and materials.” 

Response: We declare that the conflict of interest declared did not alter our adherence to all PLOS one policies. 

 

Additional Editor Comments:

Comment 1: It is essential to correct grammatical errors and typos.

Response: Done

Comment 2: To clarify whether the same households in kebeles selected/recruited in three rounds of surveys and whether there was any replacement strategy used for vacant households.

Response: The information is now provided in the “sampling procedure” sub-section (Page 10)

Comment 3: The authors need not change the use of verbal consent to written informed consent as suggested by the reviewer. Oral informed consent is acceptable for those with low literacy.

Response: OK

Comment 4: In the discussion section, the possible impact of demand creation activities and uninterrupted medical supplies system should be added with appropriate citation rather than included in the results section as indicated by the reviewer:

Response: Corrected (Page 22, Paragraph 2)

Comment 5: " It will be good to give comparison of change of IEC activities before and after the programme, the medicine supply chain".

Response: It is true that comparison of changes in IEC activities and medical supply would be interesting and add value to the study. Unfortunately, we did not collect quantitative data on these parameters. 

Comment 6: If the positive findings are likely to be due to " maturation effect" please rewrite the conclusion.

Response: As the study is not a controlled trial, the changes in care seeking might also be explained by “Maturation effect”. The same is now stated in the Discussion and Conclusion sections. 

 

Reviewers' comments:

Reviewer I

Comment 1: Title: It is unclear why the authors have used the word ‘emerging” for region. 

Response: In Ethiopia context, regions are broadly classified: as emerging and major regions. Emerging regions have relatively lower economic status and limited access to social services. In order to clarify the issue further for international readers, we have added few sentences under the sub-section “study setting” (Page 7). 

Comment 2: There is no mention about registry of the trial, hope it has been registered.

Response: In fact, the study was a program evaluation, not a trial. It has not been registered. 

Comment 3: Background: The ICCM programme was implemented in Ethiopia in 2012, but it is unclear what was the implementation status before the research team implemented the programme.

Response: the following sentence “Even though the iCCM program started to be implemented in Ethiopia earlier, the program was rudimentary in Benishangul Gumuz region until 2017.” Is now added in the “Background” section, Paragraph 6 (Page 5). 

Comment 4: Methods: Authors have used a pre and post design and the aim is to evaluate the change in care seeking behaviours after implementation of the programme not the associations? the objectives need to be clearly stated.

Response: The objective is now restated (Page 7). 

Comment 5: Intervention: The authors need to describe the intervention in more details, what was missing, what was put in place by the research team. How was health system strengthening done.

Response: The required information is now given under the sub-section “Description of the intervention” (Page 8).

Comment 6: IEC: what IEC activities were done specific to ICCM programme, its frequency.

Response: Further clarification is now given under the sub-section “Description of the intervention” (Page 8).

Comment 7: Supervision: More details about the supervision, what was done, what was the rate of retention of skills will be valuable to add. 

Response: The point raised by the reviewer is important. Unfortunately, we did not measure the rate of retention of skills. 

Comment 8: Sampling procedure: It is unclear how the households were selected for the three surveys, were the same kebeles selected. 

Response: Once the village was selected, a rapid listing of all eligible children in the village was made, and eligible children were selected randomly. Please note that at the final sampling stage we selected, eligible children, not households. Household selection was not made to reduce the number of sampling stages. During each survey round, the same kebele and village were studied; however, study subjects were selected independently. We have now added further clarification to this section (Page 10). 

Comment 9: Ethical consideration: Although the ethics committee has cleared the project, a witnessed written informed consent could have been obtained. 

Response: Yes, it is true but it depends on the decision of the IRB. The IRB sometimes may recommend for such arrangements. But in our case, in line with the direction of the IRB that approved the proposal, independent verbal consent was secured. 

Comment 10: Data Analysis: The rationale of selecting variables for adjustment has not been described. The authors need to give definition of the wealth index if that is consistent with the definition used in national surveys.

Response: The required information is now given in the data analysis section (Page 12). 

Comment 11: Tables and Results: It will be good to give a trial profile of the exclusions, the authors have not mentioned children with what illness were selected.

Response: A section on “Morbidity pattern” and a new table (Table 2) are now added (Page 16). 

Comment 12: Table ! there is typographical error endline survey has been labelled as midline.

Response: Sorry for this silly error. Corrected. 

Comment 13: The programme was implemented for 2-59 months, however the mean age of children who were included in survey is 27 months, it is unclear whether the younger children did not participate or did not have any illness.

Response: As shown in Table 1, children 2-11 months were underrepresented, contributing only to about 5% of the total sample size. This was because, infants 0-11 months were selected for another parallel “Community-based newborn “(CBNC)” survey. This may limit the generalizability of the findings to infants. The same is now stated among the limitation of the study (Last paragraph of the Discussion section, Page 24).

Comment 14: It will be good to give comparison of change of IEC activities before and after the programme, the medicine supply chain.

Response: The comments of the reviewer are valid. However, we did not collect quantitative data on these parameters. 

 

Reviewer #2

Comment 1: Line 84: There may be a typographical error, “A multicountry study that included ……”

Response: Thank you. Corrected. 

Comment 2: Line 91: What is meaning of “revitalized the iCCM program …”? Is there previous history of implementing iCCM in the area? If yes, how is it different from previous iCCM program?

Response: The required information is now provided under the section “Description of the intervention”. The implementation of the iCCM program was started in Benishangul Gumuz region in 2014. However, the earlier program was rudimentary and did not bring meaningful change at the group level due to multiple problems including weak health extension program, limited coverage, shortage of human resources for health and turnover of trained health workers. The current program implemented the program according to the national iCCM protocol but included system strengthening and stronger community mobilisation components. 

Comment 3: Line 148 & 150: What are references for design effect 1.5 and intra-cluster correlation 2%?

The DEFF of 1.5 was calculated using standard formula using ICC of 2%. The reference for the ICC of 2% is now added (Reference # 30). 

Comment 4: Comment 4: Line 168: The author mentioned that care-seeking was defined as any of the individual conditions the child had during two weeks. But did not explain clearly how to determine when two or more signs and symptoms (e.g. fever with cough and diarrhea) were occurred in the single time of illness. In my opinion, the definition of care-seeking practice should be whether they visited to health facility or health post for every single time of illness. Reporting number of care seeking practice based on individual signs and symptoms (S & S) would lead estimation bias because one or more S & S can be occurred during single time of illness.

Response: The point raised by the reviewer is correct. When the children had two distinct illness in the refence period (as judged by the caregiver) both conditions were considered as different illnesses. However, as the reviewer noted, when the child had multiple symptoms of the same illness, we only considered the chief complaint of the mother. Clarification is now given under the section “Variables of the study” (Page 10).

Comment 5: Line 182: How the care-seeking practice was assessed? Is it record review or reported care-seeking practice by asking questionnaire?

Response: Care-seeking practice was assessed based on the reports of the caregiver without reviewing any formal medical records. The same is now stated under the “Data collection” sub-section (Page 11). 

Comment 6: Line 191: What is reference for 13 danger signs?

Response: Reference # 31 is now added.

Comment 7: In Table 1, the 4th column heading should be “Endline”.

Response: Corrected.

Comment 8: There is same typographical error in the paragraph interpreting table 4 and footnote of table 4, which should be “one way walking distance…”.

Response: Sorry for the silly error. Corrected.

---

## [Decision Letter · Decision Letter 1]

15 Oct 2020

PONE-D-20-26348R1

Changes in care-seeking for common childhood illnesses in the context of Integrated Community Case Management (iCCM) implementation in an emerging region of Ethiopia

PLOS ONE

Dear Dr. Gebremedhin,

Thank you for submitting your manuscript to PLOS ONE. After careful consideration, we feel that it has merit but does not fully meet PLOS ONE’s publication criteria as it currently stands. Therefore, we invite you to submit a revised version of the manuscript that addresses the points raised during the review process.<please by="" manuscript="" revised="" submit="" your="">

Please include the following items when submitting your revised manuscript:</please>

We look forward to receiving your revised manuscript.

Kind regards,

Khin Thet Wai, MBBS, MPH, MA (Population & Family Planning Resear

Academic Editor

PLOS ONE

Additional Editor Comments (if provided):

English language correction is deemed necessary.

Reviewers' comments:

Reviewer's Responses to Questions

**Comments to the Author**

1. If the authors have adequately addressed your comments raised in a previous round of review and you feel that this manuscript is now acceptable for publication, you may indicate that here to bypass the “Comments to the Author” section, enter your conflict of interest statement in the “Confidential to Editor” section, and submit your "Accept" recommendation.

Reviewer #1: (No Response)

2. Is the manuscript technically sound, and do the data support the conclusions?

Reviewer #1: Yes

3. Has the statistical analysis been performed appropriately and rigorously? 

Reviewer #1: Yes

4. Have the authors made all data underlying the findings in their manuscript fully available?

Reviewer #1: No

5. Is the manuscript presented in an intelligible fashion and written in standard English?

Reviewer #1: No

6. Review Comments to the Author

Reviewer #1: All the comments have not been adequately addressed

Previous Comment 1:

The authors have explained the meaning of émerging’ but the ‘word’ does not make sense in the title.

Previous Comment 2:

Unsure if Ethiopia system does not need the observational studies to be registered. For most registries all types of research need to be registered.

Previous Comment 5

Description of the intervention has been added still unclear what support other than medicines and improving linkages was done by research team It is unclear if was supervision was by the health system.

Previous Comment 7

Has this been added as the limitation of the trial.

Previous Comment 8

Is there a possibly if the same child could have been selected in all 3 surveys. If yes, would that primed the caregivers with the responses to be given.

Previous Comment 11

Trial profile is still missing.

Previous Comment 13

Not added in limitation as mentioned by author.

Previous Comment 14

Should add as a limitation.

7. PLOS authors have the option to publish the peer review history of their article (what does this mean?). If published, this will include your full peer review and any attached files.

Reviewer #1: No

---

## [Author Response · Author response to Decision Letter 1]

24 Oct 2020

Comment from the editor: English language correction is deemed necessary.

The manuscript is now edited further. 

Comments of Reviewer 1

Comment 1: The authors have not fully made all data underlying the findings. 

Response: We disagree with the evaluation of the reviewer. We have already made our data available as a supporting table (S2 table). 

Comment 2: Previous Comment 1: The authors have explained the meaning of ‘emerging’ but the ‘word’ does not make sense in the title.

Response: The title is now modified accordingly. 

Comment 3: Previous Comment 2: Unsure if Ethiopia system does not need the observational studies to be registered. For most registries all types of research need to be registered.

Response: As the reviewer mentioned, observational studies can also be registered but we don’t think that’s mandatory. That’s why the study was not registered. 

Comment 4: Previous Comment 5: Description of the intervention has been added still unclear what support other than medicines and improving linkages was done by research team It is unclear if was supervision was by the health system.

Response: Please note that this is an observational study so that the researchers were not directly involved in the implementation of the program. But if the concern of the reviewer is on the role Emory University in the implementation of iCCM program, the following activities were implemented by Emory University: training of frontline health workers, consolidating referral linkage between health posts and health centers, instating quality improvement framework at various levels of the system, provision of supervisory skill training for health extension worker supervisors, joint supportive supervision and monthly couching of health extension workers. We have now made some correction on the section “Description of the intervention” to make the issue clear for readers. 

Comment 5: Previous Comment 7: Has this been added as the limitation of the trial.

Response: this is now listed as the limitation in the last paragraph of the discussion section (Page 24)

Comment 6: Previous Comment 8: Is there a possibly if the same child could have been selected in all 3 surveys. If yes, would that primed the caregivers with the responses to be given.

Response: It would be very unlikely for a child to be included in multiple surveys for two reasons: (1) the chance a child would be sick during two or more survey rounds is obviously low, (2) in each survey round a random and independent sample of children were selected. So we don’t think the concern of the reviewer is a major concern in our study 

Comment 7: Previous Comment 11: Trial profile is still missing.

Response: It would be more interesting to have a flow chart that describe the characteristics of the study subjects excluded from the study, unfortunately we did not collect data on subjects excluded from the study. 

Comment 8: Previous Comment 13: Not added in limitation as mentioned by author.

Response: We disagree with the evaluation reviewer. As you may read from the last paragraph of the discussion section, we have discussed the issue as a limitation “Furthermore, infants 2-11 months were underrepresented (contributed to only 5-6% of the total children enrolled) which may limit the generalizability of the findings of the study.”

Comment 9: Previous Comment 14: Should add as a limitation.

Response: We have now stated this as a limitation in the last sentence of the discussion section (Page 24).

---

## [Editor Report · Decision Letter 2]

3 Nov 2020

Changes in care-seeking for common childhood illnesses in the context of Integrated Community Case Management (iCCM) program implementation in Benishangul Gumuz region of Ethiopia

PONE-D-20-26348R2

Dear Dr. Gebremedhin,

We’re pleased to inform you that your manuscript has been judged scientifically suitable for publication and will be formally accepted for publication once it meets all outstanding technical requirements.

Kind regards,

Khin Thet Wai, MBBS, MPH, MA (Population & Family Planning Resear

Academic Editor

PLOS ONE

Additional Editor Comments (optional):

All comments of reviewers are fully addressed.
---

## [Editor Report · Acceptance letter]

5 Nov 2020

PONE-D-20-26348R2 

Changes in care-seeking for common childhood illnesses in the context of Integrated Community Case Management (iCCM) program implementation in Benishangul Gumuz region of Ethiopia 

Dear Dr. Gebremedhin:

I'm pleased to inform you that your manuscript has been deemed suitable for publication in PLOS ONE. Congratulations! Your manuscript is now with our production department. 

Kind regards, 

on behalf of

Dr. Khin Thet Wai 

Academic Editor

PLOS ONE